# Epigenetic Silencing of LMX1A Contributes to Cancer Progression in Lung Cancer Cells

**DOI:** 10.3390/ijms21155425

**Published:** 2020-07-30

**Authors:** Ti-Hui Wu, Shan-Yueh Chang, Yu-Lueng Shih, Chih-Feng Chian, Hung Chang, Ya-Wen Lin

**Affiliations:** 1Graduate Institute of Medical Sciences, National Defense Medical Center, Taipei 11490, Taiwan; chestsurgerytsgh@gmail.com (T.-H.W.); leehornwok@gmail.com (S.-Y.C.); albreb@ms28.hinet.net (Y.-L.S.); 2Division of Thoracic Surgery, Department of Surgery, Tri-Service General Hospital, National Defense Medical Center, Taipei 11490, Taiwan; hung@ndmctsgh.edu.tw; 3Division of Pulmonary and Critical Care, Department of Internal Medicine, Tri-Service General Hospital, National Defense Medical Center, Taipei 11490, Taiwan; sonice3982@gmail.com; 4Division of Gastroenterology, Department of Internal Medicine, Tri-Service General Hospital, National Defense Medical Center, Taipei 11490, Taiwan; 5Department and Graduate Institute of Microbiology and Immunology, National Defense Medical Center, Taipei 11490, Taiwan; 6Graduate Institute of Life Sciences, National Defense Medical Center, Taipei 11490, Taiwan

**Keywords:** *LMX1A*, DNA methylation, lung cancer, tumor suppressor, epithelial mesenchymal transition

## Abstract

Epigenetic modification is considered a major mechanism of the inactivation of tumor suppressor genes that finally contributes to carcinogenesis. LIM homeobox transcription factor 1α (*LMX1A*) is one of the LIM-homeobox-containing genes that is a critical regulator of growth and differentiation. Recently, *LMX1A* was shown to be hypermethylated and functioned as a tumor suppressor in cervical cancer, ovarian cancer, and gastric cancer. However, its role in lung cancer has not yet been clarified. In this study, we used public databases, methylation-specific PCR (MSP), reverse transcription PCR (RT-PCR), and bisulfite genomic sequencing to show that *LMX1A* was downregulated or silenced due to promoter hypermethylation in lung cancers. Treatment of lung cancer cells with the demethylating agent 5-aza-2’-deoxycytidine restored *LMX1A* expression. In the lung cancer cell lines H23 and H1299, overexpression of LMX1A did not affect cell proliferation but suppressed colony formation and invasion. These suppressive effects were reversed after inhibition of LMX1A expression in an inducible expression system in H23 cells. The quantitative RT-PCR (qRT-PCR) data showed that LMX1A could modulate epithelial mesenchymal transition (EMT) through *E-cadherin (CDH1*) and *fibronectin* (*FN1)*. NanoString gene expression analysis revealed that all aberrantly expressed genes were associated with processes related to cancer progression, including angiogenesis, extracellular matrix (ECM) remodeling, EMT, cancer metastasis, and hypoxia-related gene expression. Taken together, these data demonstrated that LMX1A is inactivated through promoter hypermethylation and functions as a tumor suppressor. Furthermore, LMX1A inhibits non-small cell lung cancer (NSCLC) cell invasion partly through modulation of EMT, angiogenesis, and ECM remodeling.

## 1. Introduction

Lung cancer is the most common cause of cancer-related death worldwide, and survival is largely dependent on the tumor stage [1,2]. Localized disease has a five-year survival rate of approximately 55%, whereas distant disease has a five-year survival rate of approximately only 4%. Unfortunately, approximately 57% of patients are diagnosed with distant disease [2]. Therefore, major efforts are being made to identify molecular markers, including aberrant DNA methylation, circulating cell-free tumor DNA, noncoding RNA, and proteomic markers, to facilitate early detection of lung cancer to improve prognosis and survival [3,4,5,6,7].

The progression of lung cancer is a complicated process, involving a series of genetic and epigenetic changes [8,9,10]. Recent research has demonstrated a promising advance in targeted therapy for lung cancers with epidermal growth factor (EGFR) mutations [11]. An increasing number of genes, such as *p16*, *RASSF1A, APC, DAPK, MGMT*, and *FHIT*, have demonstrated abnormal methylation in lung cancers [12,13,14,15,16,17]. In addition to genetic mutations, aberrant DNA methylation is another mechanism for inactivation of tumor suppressor genes to promote the occurrence and progression of lung cancer [18,19,20]. The identification of genetic mutations and epigenetic changes will further improve biomarker-driven precision treatment for lung cancer patients.

Homeobox genes include a large family of developmental regulators that are crucial for growth and differentiation. LIM homeobox genes, which encode LIM-homeodomain (LIM-HD) proteins, are one of the most important subfamilies of homeobox genes. They have two LIM domains in the amino termini and an HD domain for interacting with specific DNA sequences. Recently, research has shown that LIM homeobox genes play an important role in cancer development [21,22,23,24,25,26,27,28,29].

The LIM homeobox transcription factor 1α (*LMX1A*) gene is one of the LIM-homeobox-containing genes [30]. LMX1A has been reported to be a critical regulator of cell fate decisions and of the differentiation of human embryonic stem cells into midbrain dopaminergic neurons [31,32]. Recently, *LMX1A* has been found to be hypermethylated in cervical cancer, gastric cancer, bladder cancer, ovarian cancer, and colorectal cancer [25,26,28,29,33]. Moreover, *LMX1A* was reported to function as a tumor suppressor in cervical, gastric, and ovarian cancers [25,28,29], and its methylation was significantly associated with recurrence in bladder cancer [26] and survival in stage I and II colorectal cancer patients [33]. However, the role of LMX1A in lung cancer remains unclear. The purpose of this study was to investigate the epigenetic regulation and biological function of LMX1A in lung cancer. Our data demonstrated that LMX1A is epigenetically regulated and functions as a tumor suppressor in non-small cell lung cancer (NSCLC).

## 2. Results

### 2.1. Promoter Methylation of LMX1A in Non-Small Cell Lung Cancer

To determine whether promoter methylation of the *LMX1A* gene is a common epigenetic event in non-small cell lung cancer, we first used DNA methylation data from the SMART App (http://www.bioinfo-zs.com/smartapp) [34] to analyze the methylation levels of *LMX1A* in 458 lung adenocarcinoma (LUAD) and 364 lung squamous cell carcinoma (LUSC) samples. As shown in Figure 1A, the average β value was higher in the LUAD group than in the normal control group (*p* < 0.0001). A similar phenomenon was observed in LUSC (*p* < 0.0001). Then, we analyzed the methylation level and mRNA expression of *LMX1A* in lung cancer cell lines (Figure 1B,C). The data showed that *LMX1A* was frequently hypermethylated and downregulated in NSCLC cells. To further confirm the methylation-specific polymerase chain reaction (MSP) results, we applied bisulfite genomic sequencing to assess the methylation level in the promoter region of *LMX1A*. The results of bisulfate genomic sequencing revealed that *LMX1A* was poorly methylated in A549 and H647 cells (<20%) and highly methylated in H1299 (65%), H358 (70%), CL1-0 (93%), and H23 (88%) cells (Figure 2 and Appendix A).

### 2.2. Correlation between Methylation Status and Gene Expression Level of LMX1A in NSCLC Cell Lines

To assess the correlation between *LMX1A* methylation and gene expression, we analyzed the messenger RNA (mRNA) expression level and DNA methylation status of *LMX1A* in eight NSCLC cell lines by quantitative reverse transcription polymerase chain reaction (qRT-PCR) and quantitative MSP (Q-MSP). The Q-MSP results (Figure 3A) showed a similar trend to the results from MSP gels and bisulfite sequencing (Figure 1 and Figure 2). The qRT-PCR data showed that *LMX1A* expression was detectable in A549 cells but undetectable in the other NSCLC cells (Figure 3A). To confirm the inverse correlation of these parameters, we treated CL1-0, H23, and HT1299 cells with a DNA methylation inhibitor (5-aza-2’-deoxycytidine; DAC). The reverse transcription polymerase chain reaction (RT-PCR) data revealed that *LMX1A* expression was restored in CL1-0, H23, and HT1299 cells after treatment with DAC (Figure 3B). In addition, methylated DNA was decreased after these cells were treated with a DNA methylation inhibitor (5-aza-2ʹ-deoxycytidine; DAC) (Figure 3B).

### 2.3. Overexpression of LMX1A Inhibits the Colony Formation and Invasion of NSCLC Cells in a Constitutive Expression System

To elucidate the role of LMX1A in lung cancer, we analyzed the A549, H1299, H23, CL1-0, CL1-3, H358, and H647 cell lines for endogenous LMX1A expression. The results showed that the LMX1A protein was undetectable in all cell lines, except for A549 cells and normal lung tissues. LMX1A expression was downr egulated in cancer cell lines compared to normal lung tissue (Appendix A). LMX1A-expressing stable clones were generated using H23 and H1299 cells, and LMX1A expression was confirmed by Western blot analysis of the V5 tag protein (Figure 4A and Appendix A). The overexpression of LMX1A did not significantly alter cell viability (Figure 4B), but it did strongly repress the colony formation, indicating suppression of cell transformation (Figure 4C). Moreover, the stable overexpression of LMX1A decreased the number of invading cells (Figure 4D). Representative images of colony formation and invasion assays are shown in Figure 4C,D.

### 2.4. LMX1A Represses the Colony Formation and Invasion of NSCLC Cells in an Inducible Expression System

To further analyze the suppressive effects of LMX1A on tumorigenesis, we established an inducible expression system in NSCLC (H23) cells. Then, we treated the cells with 1 μg/mL doxycycline (Dox) to induce LMX1A expression. Western blotting analysis confirmed that Dox treatment increased the LMX1A protein levels (Figure 5A and Appendix A). We assessed the effects of LMX1A overexpression on cell growth, colony formation, and cell invasion after induction for 7 days. Although the viability of the LMX1A-overexpressing cells was not notably different from that of the controls (Figure 5B), LMX1A overexpression repressed colony formation (Figure 5C) and invasion (Figure 5D). These results were consistent with the data obtained using cells stably expressing LMX1A. In addition, the suppressive effects of LMX1A on colony formation and invasion were reversed after knockdown of LMX1A expression by withdrawal of Dox. Taken together, these data demonstrated that LMX1A suppresses cell transformation and the invasive phenotype.

### 2.5. LMX1A Inhibits NSCLC Cell Invasion Partly through Modulation of Epithelial Mesenchymal Transition (EMT)

The epithelial mesenchymal transition (EMT) is a crucial developmental process that is often activated during cancer invasion and metastasis [35]. In a previous study, we demonstrated that LMX1A could suppress cervical cancer invasion through an incomplete EMT [29]. Here, we assessed whether LMX1A inhibited cancer invasion through EMT using qRT-PCR in H23 cells (Figure 6). Our data showed that LMX1A increased epithelial markers such as *E-cadherin* (*CDH1*) and decreased the mesenchymal markers *fibronectin* (*FN1*). Knockdown of LMX1A by withdrawal of doxycycline reversed *CDH1* expression but not *FN1* expression. The expression of *N-cadherin* (*CDH2*) remained high despite the overexpression of LMX1A. EMT-related transcription factors, such as *SNAIL*, *SLUG*, and *TWIST,* were not affected by overexpression of LMX1A (Appendix A). The protein expression of Twist and Snail was not altered with manipulation of LMX1A (Appendix A). Moreover, restoration of LMX1A significantly increased the expression of *inhibitor of DNA-binding gene 2* (*ID2)* in H23 cells. The protein results of E-cadherin (CDH1), N-cadherin (CDH2), and inhibitor of DNA-binding gene 2 (ID2) were similar to the RNA expression data (Appendix A). These results suggest that LMX1A suppresses NSCLC invasion partly through EMT-related mechanisms. To further explore the potential mechanism responsible for LMX1A suppression of cancer cell invasion, we performed a NanoString gene expression analysis to investigate differentially expressed genes related to cancer progression upon LMX1A overexpression. Differentially expressed genes in the *LMX1A* -V5 H23 cell line are summarized in Table 1 and Appendix A. Fourteen genes, including *SRGN, SCG2, HGF, AMH*, and *FN1*, were downregulated (fold change < –2). However, only three genes (*THBS1*, *LEFTY1*, and *ID2*) were upregulated (fold change > 2). LMX1A could repress four genes that belong to mesenchymal (Mes) cell markers (Table 1). All of the aberrantly expressed genes were related to processes involved in cancer progression, including angiogenesis, extracellular matrix (ECM) remodeling, cancer invasion, and cancer metastasis. These data suggested that LMX1A inhibits NSCLC cell invasion partly through modulation of EMT, angiogenesis, ECM remodeling, and hypoxia-related genes.

## 3. Discussion

Lung carcinogenesis involves a multistage stepwise process with the accumulation of genetic and epigenetic alterations [36]. Epigenetic alterations include DNA methylation, histone modifications, and noncoding RNA expression [20]. Among them, aberrant DNA methylation-induced silencing of tumor suppressor genes is a hallmark and an early event in lung carcinogenesis [37,38]. Multiple genes, such as *CDKN2A, DAPK, MGMT, RARB, RASSF1A*, and *TERT*, have been found to be methylated early in lung premalignant lesions, and the level of methylation increases with tumor progression from premalignant lesions to neoplasms [39,40,41,42,43]. These methylated genes have crucial functions in cell cycle control, apoptotic signaling, DNA repair, cell growth, and immortalization. In this study, we found that *LMX1A* was methylation-silenced in non-small cell lung cancer (Figure 1, Figure 2 and Figure 3 and Appendix A). In addition, LMX1A served crucial functions in tumor invasion and epithelial-mesenchymal transition (Table 1). To our knowledge, there are no data reporting epigenetic regulation of *LMX1A* expression and its function in lung cancer.

DNA methylation is not only important in lung carcinogenesis but also serves as a biomarker for early lung cancer detection [7,44,45,46,47,48,49]. We analyzed the data from The Cancer Genome Atlas (TCGA) database through the UALCAN (http://ualcan.path.uab.edu) web portal and showed that methylation of *LMX1A* in non-small cell lung cancer occurred early in stage I patients and was then maintained throughout all the stages of non-small cell lung cancer (Appendix A) [50]. This result implied that *LMX1A* methylation could be used as a diagnostic biomarker for the early detection of lung cancer.

*LMX1A*, as a transcription factor, is one of the LIM-homeobox genes that plays a crucial role during development [22]. Recently, LMX1A has also been reported to play an important role in cancer. Our research group has reported that *LMX1A* was methylation-silenced and that the expression of LIMX1A inhibited colony formation and invasion in cervical cancer [29]. We also found that *LMX1A*, in combination with *NKX6.1*, *SOX1*, and *ZNF177*, could serve as a stage-independent prognostic marker in colorectal cancer [33]. In ovarian cancers, *LMX1A* was reported to be hypermethylated, and its expression inhibited cell proliferation, migration, invasion, and colony formation [25]. In addition, Zhao and his colleagues found that methylation of *LMX1A* could be used as an early diagnostic biomarker and was associated with recurrence in bladder cancer [26]. Dong and his colleagues found that methylation-mediated inactivation of LMX1A and restoration of LMX1A induced cell apoptosis and suppressed anchorage-independent growth in gastric cancer [28]. All of the above findings identify LMX1A as a tumor suppressor. However, LMX1A did not function as a tumor suppressor in other types of cancer. Tsai and his colleagues reported that the higher expression of LMX1A shown by immunohistochemical staining correlated with the WHO grade of meningioma and glioma [27]. Tsai and his colleagues reported that the higher expression of LMX1A was strongly correlated with the histologic grade and pathologic stage of pancreatic ductal adenocarcinomas [23]. LMX1A functions as an oncogene instead in these cancers.

In this study, we demonstrated that promoter methylation of *LMX1A* was frequent in NSCLC cell lines and tumor tissues (Figure 1, Figure 2 and Figure 3 and Appendix A). The methylation-induced silencing of *LMX1A* could be reversed after cells were treated with a DNA methylation inhibitor in the three cell lines (Figure 3B). Indeed, when we treated H647 cells with a histone deacetylase inhibitor (trichostatin A (TSA)), LMX1A expression was restored (Appendix A). These data suggested that histone modifications are also involved in the epigenetic silencing of LMX1A in lung cancer cells. Overexpression of LMX1A did not affect cell proliferation but inhibited colony formation and invasion. These results were similar to the suppressive effect of LMX1A in cervical cancer [29].

Activation of the EMT program is a well-known critical mechanism for the acquisition of invasiveness and metastasis in cancers [35]. Chao and his colleagues reported that LMX1A could inhibit the EMT pathway by upregulating the epithelial marker *CDH1* and downregulating the mesenchymal markers *CDH2* and *FN1* by repressing the EMT-related transcription factors *SNAIL*, *SLUG*, and *SIP1* in ovarian cancer [25]. In a previous study, we showed that LMX1A could suppress the EMT pathway partially by inhibition of the EMT-related transcription factors *SNAIL*, *SLUG*, and *TWIST* in cervical cancer [29]. In this study, we found that LMX1A suppressed NSCLC invasion partly through an EMT-related mechanism without influencing the EMT-related transcription factors *SNAIL*, *SLUG*, and *TWIST* (Figure 2 and Figure 6 and Appendix A). This finding suggests two possibilities. First, the regulation of *SNAIL*, *SLUG*, and *TWIST* might occur through post-translational modification instead of transcriptional regulation. Previous reports showed that no permanent genetic alterations in most EMT transcription factors were found in cancer, but transient changes were detected. These post-translational modifications or epigenetic modulation of EMT transcription factors could directly regulate their function by affecting protein stability, nuclear localization, protein–protein interactions, and ubiquitination. EMT transcription factors can drive tumor cell switching between epithelial traits for tumorigenic proliferation and a mesenchymal phenotype for invasion and migration. These transcription factors exert their function in a more flexible and reversible manner, [51,52]. Moreover, growing evidence suggests the presence of a transitional status between epithelial and mesenchymal phenotypes, i.e., the “hybrid epithelial-mesenchymal (hybrid E/M)” condition, which provides a possible reason for these conflicting results. Second, the effect on EMT was mediated through pathways other than *SNAIL*, *SLUG*, and *TWIST*. This hypothesis is supported by the finding that the EMT regulators *SRGN*, *HGF*, *FN1*, *CSF2RB*, *CKMT1A*, and *CXCR4* were downregulated in the *LMX1A*-overexpressing H23 cell lines (Table 1). All the mentioned genes have been reported to be EMT regulators and play important roles in the migration, invasion, sphere formation, or drug resistance of cancer cells [53,54,55,56,57,58,59,60,61,62,63,64,65,66,67,68,69,70,71]. All together, these data suggested that LMX1A functions as a tumor suppressor partly by regulating EMT-related genes.

LMX1A has been reported to affect pathways other than EMT. Qian and his colleagues demonstrated that LMX1A represses *C-MYC* expression by activating *ANGPTL4* to function as a tumor suppressor in gastric cancer [21]. In lung cancer, *C-MYC* is frequently dysregulated, important for cancer stem cell (CSC) properties, and associated with unfavorable patient survival [72], and its expression was significantly correlated with programmed death-ligand 1 (PD-L1) expression in NSCLC [73]. However, in the Nanostring gene expression analysis, *angiopoietin-like 4* (*ANGPL4*) and *C-MYC* were not affected after overexpression of LMX1A in H23 cells. After overexpression of LMX1A in H23 cells, ID2 protein levels were increased. Knockdown of LMX1A by doxycycline (DOX) depletion reversed the ID2 protein level. Moreover, using the Eukaryotic Promoter Database (EPD) (http://epd.vital-it.ch) , we discovered four putative homeobox binding sites in the promoter of 2.5 kb of the *ID2* gene. These data support ID2 as a potential target of LMX1A. The TGF-ß pathway induces EMT transition through many downstream signaling pathways. ID2 was significantly decreased by TGF-ß, and expression of ID2 was restored after treatment with a TGF-ß inhibitor [74]. It has been reported that ID2 inhibited cancer invasion and metastasis [75], and the mRNA expression levels of ID2 were inversely correlated with the tumor metastasis stage in clinical samples [76]. Moreover, transient overexpression of ID2 in the parental H23 cell line also represses cancer invasion (Appendix A). Taken together, these data provide evidence to further elucidate how LMX1A regulates ID2, and this is linked with its function.

In this study, we evaluated the role and function of LMX1A only in human NSCLC cells. Based on these in vitro studies, a xenograft animal model should be established to clarify LMX1A’s tumor suppressor role in NSCLC in vivo. To further explore the downstream signaling pathways related to the tumor suppressor function of LMX1A, researchers should use RNA sequencing technology for comprehensive analysis. Our results need further investigations in future studies.

In brief, promoter methylation leads to the silencing of LMX1A and contributes to cancer progression. LMX1A inhibits NSCLC cell invasion partly through modulation of EMT, angiogenesis, and ECM remodeling.

## 4. Materials and Methods

### 4.1. SMART (Shiny Methylation Analysis Resource Tool) App

SMART (Shiny Methylation Analysis Resource Tool) App (http://www.bioin fo-zs.com/smart app) is a web application for analyzing the DNA methylation of human cancers [34]. In this study, we used DNA methylation data in the SMART App for analysis of the methylation levels of *LMX1A* in 41 tissue samples from normal controls and 458 tissue samples from lung adenocarcinoma (LUAD) patients; 364 tissue samples from lung squamous cell carcinoma (LUSC) patients are shown.

### 4.2. Cell Lines

A total of eight human lung cancer cell lines (A549, H1437, H23, CL1-0, CL1-3, H358, H1299, and H647) were used in this study. They were obtained from Professor Yi-Ching Wang (National Cheng Kung University, Taiwan). We cultured cells in Roswell Park Memorial Institute (*RPMI)* 1640 medium (Gibco, Gaithersburg, MD, USA) supplemented with 10% heat-inactivated fetal bovine serum (FBS) and other components as previously described [77]. All cell cultures were maintained in an incubator containing 5% CO_2_ at 37 °C. Detailed descriptions are available in the Appendix A.

### 4.3. DNA Methylation and Gene Expression Analysis

Genomic DNA of NSCLC cell lines was extracted and bisulfite-converted as previously described [78]. CpG-methylated human genomic DNA (Thermo Fisher Scientific, San Diego, CA, USA) and DNA extracted from normal peripheral blood lymphocytes were modified by sodium bisulfite to generate positive and negative controls, respectively. MSP, bisulfate sequencing, and Q-MSP were performed as previously described. For Q-MSP, the DNA methylation levels were assessed by determining the methylation index (MI) using the following formula: 100 × 2^−^^[(Cp^ °^f Gene) − (Cp^ °^f COL2A)]^. For gene expression analysis, RNA extraction and reverse transcription polymerase chain reaction (RT-PCR) were conducted as previously described [78]. The primer sequences were described previously [79] and are shown in Appendix A. Detailed descriptions are available in the Appendix A and Methods.

### 4.4. Plasmids Construction

The full-length *LMX1A* open-reading frame (ORF) was cloned into the pcDNA3.1-V5-His-TOPO constitutive expression vector (Invitrogen, Waltham, MA, USA) (termed pcDNA3.1-LMX1A) or the inducible expression vector pAS4.1w. Pbsd-aOn (Appendix A) (purchased from National RNAi Core Facility of Taiwan) (termed pAS4.1-LMX1A).

### 4.5. Generation of Cells Overexpressing Stable Clones or Inducible Expression Stable Clones

Assays for the generation of cells overexpressing stable clones or inducible expression stable clones were conducted as described previously [79]. Detailed descriptions are available in Appendix A and methods.

### 4.6. Assays for Western Blot, Cell Viability, Anchorage-Independent Growth, and Invasion

Assays for Western blot, cell viability by MTS(3-(4,5-dimethylthiazol-2-yl)-5-(3-carboxymethoxyphenyl)-2-(4-sulfophenyl)-2H-tetrazolium, Inner Salt), anchorage-independent growth, and invasion were conducted as described previously [29].

### 4.7. NanoString Gene Expression Analysis

Total RNA was isolated from NSCLC cells using TRIzol reagent (Invitrogen) according to the manufacturer’s handbook. Purity and quantity were assessed by the NanoDrop 8000 Spectrophotometer (Thermo Fisher Scientific, Waltham, MA, USA). NanoString gene expression analysis was performed on total RNA from isolated cultured cells using the nCounter Human PanCancer Progression Panel (NanoString Technologies, Seattle, WA, USA) according to the manufacturer’s instructions [80,81,82]. Briefly, 200 ng of total RNA was hybridized with reporter and capture probes in a thermocycler (65 °C) for 24 h. Samples were then loaded onto the nCounter Cartridge by using the nCounter Prep Station, and RNA was quantified by using the Digital Analyzer. After quality control analysis, normalized RNA counts were generated by negative control subtraction and the geometric mean of the housekeeping genes *TUBB, PGK1, GUSB, HPRT1, CLTC,* and *GAPDH* with nSolver Analysis Software v2.5. Genes were considered not expressed if the normalized RNA counts were less than twice the SD of the negative control counts in both the LMX1A-V5 cells and the control cells.

### 4.8. Statistical Analysis

Statistical analyses were performed using GraphPad Prism software (version 5; GraphPad Software, La Jolla, CA, USA) and SPSS software (IBM SPSS Statistics 21; Asia Analytics Taiwan, Taipei, Taiwan). All values are expressed as the mean ± SEM. The Mann–Whitney U test was used to determine differences between gene methylation levels and disease status. The Student’s *t*-test and Mann–Whitney test were used to compare colony number, cell invasion, and relative RNA expression in the different stable transfectants.

## 5. Conclusions

In summary, the current study demonstrated that LMX1A functions as a tumor suppressor partly by repressing EMT, angiogenesis, and ECM remodeling in lung cancer. Epigenetic inactivation of LMX1A abolishes the tumor suppressor function. Our data identified a methylation biomarker for lung cancer progression and suggest a potential combination of epigenetic drugs in future therapeutic approaches. 

## Figures and Tables

**Figure 1 ijms-21-05425-f001:**
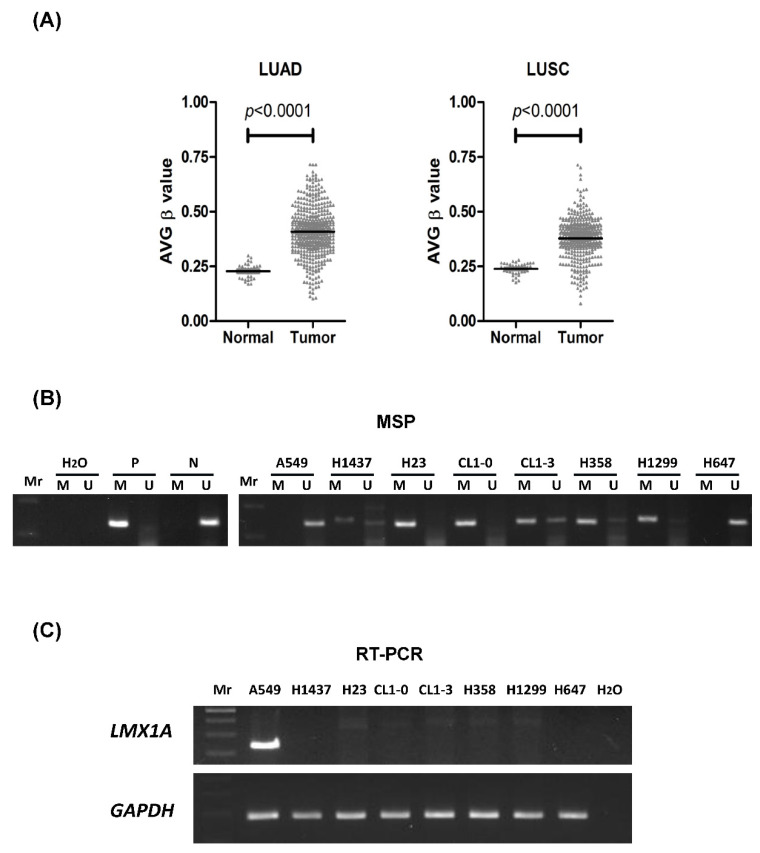
Promoter methylation levels of *LMX1A* in non-small cell lung cancer (NSCLC). (**A**) DNA methylation array data for *LMX1A* in 41 tissue samples from healthy individuals, 458 tissue samples from lung adenocarcinoma (LUAD) patients, and 364 tissue samples from lung squamous cell carcinoma (LUSC) patients from the Smart App (http://www.bioinfo-zs.com/smartapp) are shown. The results are shown as average (AVG) β values for the probes. Black lines indicate the mean AVG β value. The *p*-values for *LMX1A* methylation levels among the groups (normal versus tumor) were determined by the Mann–Whitney U test. (**B**) The promoter methylation statuses of *LMX1A* in eight NSCLC cell lines were analyzed by methylation-specific PCR (MSP) with methylated and unmethylated-specific primers. P: Positive control; N: Negative control. (**C**) Gene expression levels of *LMX1A* and the internal reference *GAPDH* in eight NSCLC cell lines were analyzed by reverse transcription polymerase chain reaction (RT-PCR). Mr: Molecular marker, or DNA ladder.

**Figure 2 ijms-21-05425-f002:**
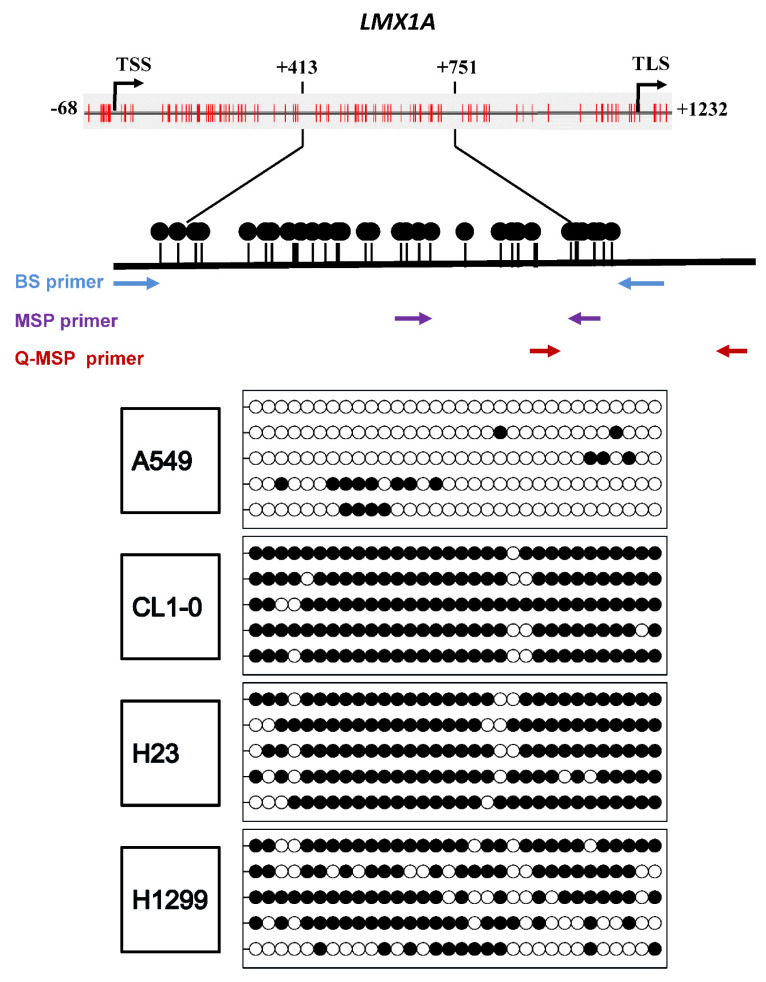
Bisulfite genomic sequencing analysis of the CpG sites located at the promoter region of *LMX1A*. The *LMX1A* methylation status in four NSCLC cell lines was analyzed by bisulfite genomic sequencing. Each clone is represented by a row, and 32 CpG sites are represented as circles. Black circles and white circles represent methylated and unmethylated CpG sites, respectively. Arrows indicate the locations of the BS primers, MSP primers, and Q-MSP. TSS: Transcriptional start site; TLS: Translational start site; BS: Bisulfate sequencing; MSP: Methylation-specific PCR; Q-MSP: quantitative methylation-specific PCR.

**Figure 3 ijms-21-05425-f003:**
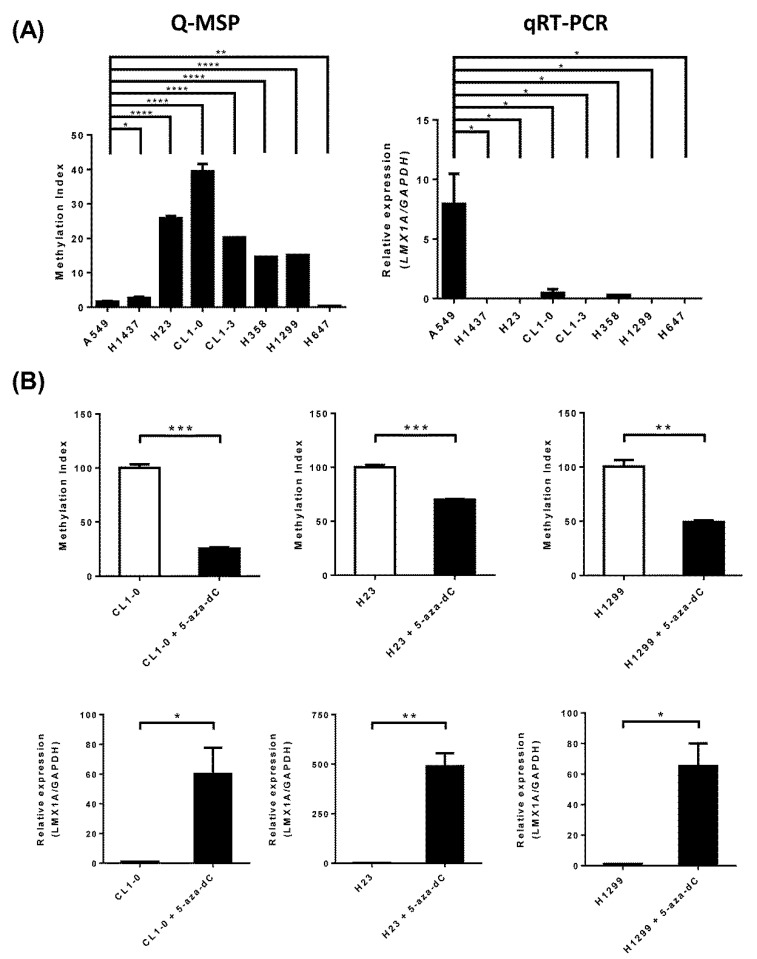
Correlation between methylation status and gene expression level of *LMX1A* in NSCLC cell lines. (**A**) Quantitative DNA methylation levels of *LMX1A* in eight NSCLC cell lines were analyzed by Q-MSP. Gene expression levels of *LMX1A* and the internal reference *GAPDH* were analyzed by quantitative RT-PCR. (**B**) Quantitative DNA methylation levels of *LMX1A* in CL1-0, H23, and H1299 cells treated with 5 µM 5-aza-2ʹ-deoxycytidine (DAC) or left untreated were determined by Q-MSP. The results are represented as differences in the methylation index (MI). Gene expression levels of *LMX1A* and the internal reference *GAPDH* in CL1-0, H23, and H1299 cells treated with 5 µM DAC or left untreated were analyzed by quantitative RT-PCR. * *p* < 0.05, ** *p* < 0.01, *** *p* < 0.001, and **** *p* < 0.0001 (Student’s *t*-test).

**Figure 4 ijms-21-05425-f004:**
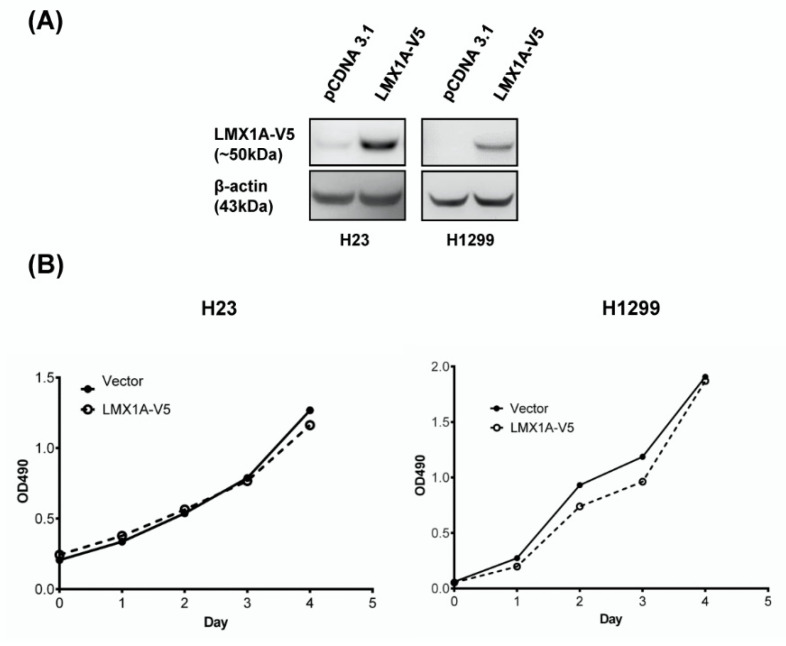
Ectopic expression of LMX1A suppresses the colony formation and invasion of cancer cells in a constitutive expression system. (**A**) Expression of LMX1A in H23 and H1299 NSCLC cell lines stably transfected with LMX1A-V5 or the empty vector (pCDNA3.1) was analyzed by Western blots using an anti-V5 antibody. β-Actin was used as an internal control. (**B**) Cell proliferation (3-(4,5-dimethylthiazol-2-yl)-5-(3-carboxymethoxyphenyl)-2-(4-sulfophenyl)-2H-tetrazolium (MTS)) assays were performed using H23 and H1299 cells expressing LMX1A-V5 or the empty vector. The absorbance values are presented as the mean ± SEM from four independent experiments. (**C**) Colony formation assays were performed using H23 and H1299 cells expressing *LMX1A*-V5 or the empty vector. (**D**) Matrigel invasion assays were performed using H23 and H1299 cells expressing LMX1A-V5 or the empty vector. These results are presented as the mean ± SEM from three independent experiments in duplicate. ** *p* < 0.01 (Mann–Whitney U test).

**Figure 5 ijms-21-05425-f005:**
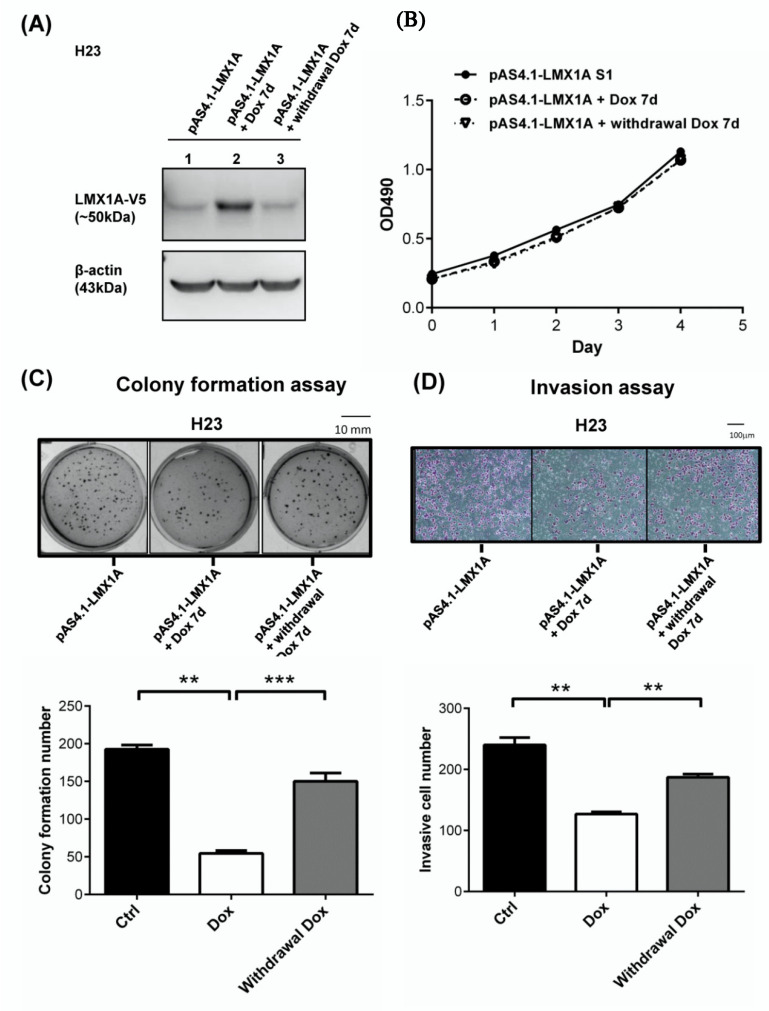
LMX1A suppresses the colony formation and invasion of cancer cells in an inducible expression system. (**A**) Doxycycline (Dox; 1 ng/μL)-inducible LMX1A expression was established in H23 cells and was assessed by Western blot analysis. β-Actin was used as an internal control. (**B**–**D**) Cells treated with Dox (1 ng/μL) for 7 days or withdrawn from Dox treatment for another 7 days were tested for cell proliferation (MTS) (**B**), colony formation (**C**), and invasion (**D**). The results are presented as the mean ± SEM from three independent experiments in duplicate. ** *p* < 0.01 and *** *p* < 0.001 (Mann–Whitney U test or Student’s *t*-test).

**Figure 6 ijms-21-05425-f006:**
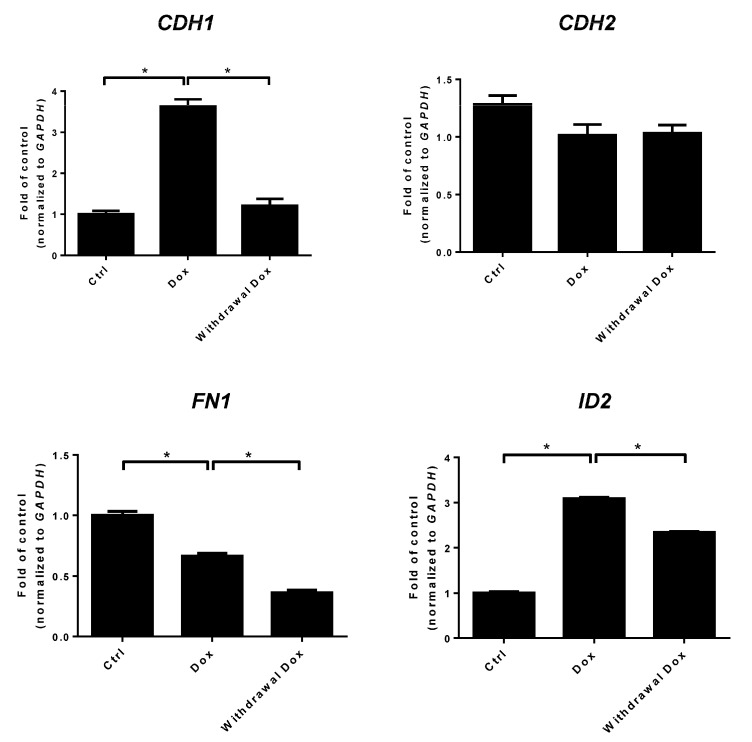
Effects of LMX1A on epithelial mesenchymal transition (EMT)-related genes in NSCLC cells. Changes in EMT-related genes (*CDH1*, *CDH2*, *FN1*, and *ID2*) after induction of LMX1A in H23 cells. Data are shown as fold changes of mRNA expression relative to control cells without doxycycline. Values are expressed as the mean ± SEM from three independent experiments, each conducted in duplicate. * *p* < 0.05 (Mann–Whitney U test or Student’s *t*-test). *CDH1: E-cadherin; CDH2: N-cadherin; FN1: Fibronectin; ID2: The inhibitor of DNA-binding gene 2*.

**Table 1 ijms-21-05425-t001:** Summary of differentially expressed genes in the LMX1A-V5 H23 cell line by NanoString gene expression analysis.

Probe Name	Accession Number	Fold Change: *LMX1A* vs. Control	*p* Value of *LMX1A* vs. Control	EMT Spectrum	Progression Categories
*SRGN*	NR_036430.1	−4.32693624	0.01084103	Mes	Angiogenesis, ECM Layers, EMT
*SCG2*	NM_003469.3	−3.88797712	0.01710246		Angiogenesis, Tumor Growth
*HGF*	NM_000601.4	−2.82349849	0.00009102		Angiogenesis, ECM Layers, EMT, Tumor Growth, Tumor Invasion
*AMH*	NM_000479.3	−2.80556583	0.04690248		Transcription Factor, Tumor Growth
*FN1*	NM_212482.1	−2.80556583	0.04547928	Mes	Angiogenesis, ECM Layers, ECM Remodeling, EMT, Tumor Invasion
*APOE*	NM_000041.2	−2.73198891	0.02757248		ECM Layers, Tumor Growth
*CSF2RB*	NM_000395.2	−2.59863329	0.0019465	Mes	ECM Layers, EMT
*MYH11*	NM_001040113.1	−2.59863329	0.0019465		Metastasis
*PIK3R6*	NM_001010855.3	−2.59863329	0.0019465		Angiogenesis
*CLU*	NM_203339.2	−2.56240726	0.02621217		Angiogenesis, ECM Layers, Tumor Growth
*CRIP2*	NM_001270837.1	−2.32546878	0.03496239		ECM Layers, Tumor Growth
*TIMP1*	NM_003254.2	−2.28131437	0.01276203		Angiogenesis, ECM Layers, ECM Remodeling, Hypoxia, Metastasis, Tumor Growth
*CKMT1A*	NM_001015001.1	−2.19464755	0.01458759		EMT
*CXCR4*	NM_003467.2	−1.99022377	0.04348437	Mes	ECM Layers, EMT, Hypoxia
*THBS1*	NM_003246.2	3.25843048	0.0395245		Angiogenesis, ECM Layers, ECM Remodeling, Hypoxia, Tumor Growth, Tumor Invasion
*LEFTY1*	NM_020997.2	2.91027427	0.04374892		Tumor Growth
*ID2*	NM_002166.4	1.99489844	0.00435035		Transcription Factor, Tumor Growth

All overexpressed genes are represented as values >1 and all underexpressed genes are represented as values <−1; Mes: Mesenchymal.

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
