# Peer review of "Epigenetic Silencing of LMX1A Contributes to Cancer Progression in Lung Cancer Cells"

_ijms, 2020, doi:10.3390/ijms21155425_

Round 1

Reviewer 1 Report

In this manuscript, the Authors evaluated the role of LMX1A in human NSCLC cells using in vitro experiments. They conclude that the methylation extent of LMX1A promoter could regulate LMX1A gene expression; moreover, they suggest that LMX1A could function as a tumor suppressor in NSCLC since the overexpression of this protein reduces NSCLC cell colony formation and invasion by modulating epithelial mesenchymal transition (EMT), angiogenesis and extracellular matrix remodeling (ECM).

The manuscript is well written, the results are well presented. ALthough, the majority of the experiments are based on gene expression analysis, the study of protein levels is rare. I agree with the authors about a possible role of LMX1A in suppressing the cell colony formation and invasion of NSCLC cells, but more experiments are required to further support the mechanisms involved these processes and the role of methylation in the regulation of LMX1A gene expression.

Major points:

  • Authors proposed an inverse correlation between LMX1A methylation and gene expression. It seems to be correct for some cells lines, such as A549, H23 or CL1-0 (Fig 3), but for in H1437 or H647 cells, the LMX1A expression is very low or absent even if the methylation level is similar to A549. How the authors could explain this point? In addition, no statistical significance is showed in the graphs of fig. 3A.
  • To exclude an aspecific effect of the DAC inhibitor on the LMX1A expression, the experiment in Fig 3B should be done with another structurally unrelated DNA methylation inhibitor. Moreover, the effect of DAC on those cells with a reduced LMX1A methylation level (A549, H1437 and H647) should be showed.
  • The experiments presented in fig. 4B-D should be performed in A549 cells by downregulating LMX1A with RNA interference technique, in order to confirm the role of LMX1A in cell proliferation, colony formation and invasion assay.
  • In fig 6, it is not clear in which cell line(s) the data presented were obtained. Moreover, the correlation between LMX1A expression and EMT markers it is not completely clear. The overexpression of LMX1A mediated by doxocycline (DOX) induced an alteration of CDH1 gene, but there is not a significance variation of CDH2 expression in the same conditions. Moreover, a reduction of FN1 is showed by using DOX, but the withdrawal of DOX induced a stronger reduction of FN1 instead of recover the FN1 level as the Ctrl sample. How the authors could explain these points? A western blot of the EMT markers analyzed in this experiment is required.
  • LMX1A expression was showed to upregulate CDH1 without influencing SNAIL, SLUG or TWIST expression, but the protein-level of these EMT-related transcription factors upon LMX1A overexpression is lacking. A western blot of SNAIL, SLUG or TWIST show be performed in an experiment similar to that of fig. 6.
  • To confirm some results suggested by the Nanostring gene expression analysis, the protein level of selected proteins among them involved in EMT or ECM remodeling should be performed.

Author Response

Dear reviewer: 

Please see the attached file, thanks.

Reviewer 2 Report

In the manuscript “Epigenetic silencing of LMX1A contributes to cancer progression in lung cancer cells” Wu and colleagues report frequent promoter hypermethylation and loss of expression of the transcription factor LMX1A in lung cancer samples from The Cancer Genome Atlas (TCGA) dataset and in lung cancer cell lines. The authors further characterize the effects of LMX1A overexpression on cell proliferation, colony formation and invasion in the lung cancer cell line H23. Finally, by Nanostring gene expression analysis they identify ID2, among other genes, to be positively regulated by LMX1A.

On the overall, the topic of the current study is surely of interest and is consistent with previous data from the literature. However, there are several concerns and limitations that refrains from supporting its publication at this point in time. Important key experiments and additional information about current experiments are missing, as detailed below, to draw conclusions.

The authors should carefully examine all points and address them in full for the manuscript to represent a contribution to the cancer field.

General comments

1/ The key finding here is that the authors describe LMX1A promoter hypermethylation in several lung cancer cell lines and in datasets from TCGA. Did the authors also analyzed TCGA lung cancer dataset for other means of LMX1A inactivation, such as deletion or mutation?

Also, is LMX1A inactivation in lung cancer mono or biallelic, in other words does LMX1A promoter hypermethylation results in loss of heterozygosity (LOH), or is LMX1A behaving as a haploinsuficient tumor suppressor?

2/ Two cell lines exhibit low methylation at the LMX1A promoter : A549 and H647, however LMX1A is expressed in A549 but not in H647. The authors should verify and comment the LMX1A gene integrity in both cell lines. If LMX1A is mutated in H647 this may be an indication why there is no expression in the absence of promoter methylation.

3/ Figure 4B and 5B: Why the authors used OD to estimate cell density of adherent cell lines (both H23 and H1299). Do the authors obtain similar results regarding cell proliferation by counting the cells?

4/ Figure 4C, 4D, 5C, 5D:  images of colonies and especially invasion are of very poor quality and should be replaced by higher magnification images, including scale bar. The invasion assay should be detailed in the methods section.

5/ A549 is the only cell line in this study that expresses LMX1A. Is there any phenotypic change upon LMX1A knock-down (by shRNA or CRISPR) in this cell line? Does this cell line harbors other alterations of the LMX1A pathway, such as LMX1A downstream effectors? In this regard, what is ID2 expression in A549, compared to other lung adenocarcinoma cells lines where LMX1A is inactivated? The authors should compare ID2 RNA and protein levels in LMX1A expressing samples (A549 as well as normal lung tissue) and inactivated (H23, H1299, H23, …cell lines). Such a comparison would be of particular interest to further strengthen the authors hypothesis and conclusions.

6/ The TetON inducible expression system used in H23 cells and relying on the pAS4.1w vector cells should be further introduced. Is this a retro or a lentiviral vector, is it integrating into the genome, is the integration site known, is the established LMX1A inducible cell line monoclonal or polyclonal? The authors refer to Supplementary methods section, which is missing.

7/ The authors propose that ID2 is a transcriptional target of LMX1A. This is an exciting hypothesis, however still preliminary in the current version of the manuscript. In order to further strengthen their interpretations, the authors should complete the manuscript by the following:

                  * Verify by western blot if ID2 levels are indeed higher in the isogenic cell lines overexpressing LMX1A, in comparison to parental the cell line inactivated for LMX1A, also see comment 5. It is crucial to validate that ID2 is present and accumulated at protein levels in order to support the authors hypothesis about a mechanistic link between LMX1A regulating EMT trough ID2 downstream effectors.

                  * Include the description of the in silico identified LMX1A putative binding sites in the results and methods section). Data analysis of available ChIPseq of LMX1A (or LMX1A ChiP performed by the authors themselves) would be an added value to further strengthen the hypothesis that LMX1A is a direct transcriptional regulator of ID2.

8/ Does overexpression of ID2 in the parental H23 cell line phenocopies LMX1A overexpression? This would be of a particular interest in supporting the actual conclusions drawn by the authors.

9/ More information is required regarding the NanoString analysis. Did the authors used a custom codeset for EMT-related pathway in the current study? A referecen for NanoString would be a plus.

Minor comments

  1. LUng ADenocarcinoma is called LUAD (as annotated in Figure 1A, as well as in the SMART App and TCGA) and not LUDA as found in the main text (line 82, 83) and Figure 1 legend (line 95).
  2. Line 80 : The date in the SMART App are from TCGA, this should be indicated for better clarity of the message.
  3. Line 120, 123 : indicate the full name for DAC (5-aza-deoxycytidine)
  4. Line 342: “and other components are previously described”. To gain into clarity, the authors are invited to cite all components relevant to the cell culture conditions relevant to the present study in the methods section of this manuscript.
  5. Line 139-140 of the main text and associated figure S2 are referring to the use of normal lung tissue. How did the authors get access to this normal lung control tissue and is it from mouse or human origin?
  6. The claim on Line 223 need to be carefully worked out. “No data reporting epigenetic regulation of LMX1A expression … in lung cancer” is in contradiction with line 227 “the analyzed data from TCGA … showed that methylation of LMX1A in NSCLC occurred early in stage I patients…etc”, citing the reference 50 of the manuscript.
  7. Figure S3 : in the legend is indicated “* p<0.05”, however none of the 3 genes tested by qPCR show difference in expression between conditions.
  8. Figure 4C, 4D, 5C, 5D : The representation method by histogram suggests an important difference in the number of colony formation and invasion (ranging approximately from 1/3 to 1/6) between conditions, however the statistical analysis of tree independent experiments give only p<0.05. This suggests that the choice of histogram representation may not reflect at best those tests. Maybe scatter plots would be a better choice of representation and a supplementary file including raw data from quantifications would be of help

Author Response

Dear Reviewer:

Please see the attached file, thanks.

Reviewer 3 Report

In this manuscript, Wu et al demonstrate that the LMX1A promoter is highly methylated in many lung cancer cell lines, affecting LMX1A transcription, colony formation and invasion. They monitor downstream gene expression changes, identifying multiple candidates involved in the epithelial to mesenchymal transition (EMT). The experiments are an extension of previous work showing that LMX1A is methylated and mis-regulated in other cancer types. The experiments presented were well described, justified, and easy to follow. The argument posed in the latter part of the discussion was difficult to follow in certain places, jumping from gene to gene (lines 253-303). This area had very long sentences that could be simplified (example: line 266-272). Overall this is an interesting manuscript providing correlative evidence that HOX gene regulation of EMT factors promotes colony formation and cell invasion in cell lines which suggests an ability to metastasize. Further experiments can be pursued to provide mechanistic insight into how LMX1A performs this function—is it via alterations in EMT genes identified here, or are there other players? These are interesting questions.

The experiments here raised a number of interesting questions that could increase significance/novelty:

1) Is LMX1A regulation of EMT the cause of the invasion and colony formation phenotypes observed? Can the M genes be expressed alone to determine effect on colony formation or invasion or both?

2) Is mis-regulation of SRGN alone sufficient to mis-regulate the EMT genes described in H1299 and H23 cells? or the invasion/colony formation?

3) Why does FN1 expression not recover after DOX removal? The argument in the discussion appears that post-translational modifications can occur…. Could the methylation of its promoter also be altered?

4) It would be really interesting for the authors to discuss potential reasons why LMX1A acts as an oncogene in some cases (line 247) and a tumor suppressor in others but that may be outside the scope of this paper.

5) LUAD not LUDA

Author Response

Dear Reviewer:

Please see the attachment file, thanks

Round 2

Reviewer 1 Report

The authors answered to the majority of the questions and modified the manuscript in accordance. I agree with the publication of the paper.

Reviewer 2 Report

The authors have addressed the majority of the raised points and the new results further strengthen or clarify the initial message of the manuscript.

A minor remark :

Add representative images of the different conditions for the experiment quantified in Figure S7B.